# The Interaction of Mechanics and the Hippo Pathway in *Drosophila melanogaster*

**DOI:** 10.3390/cancers15194840

**Published:** 2023-10-03

**Authors:** Jia Gou, Tianhao Zhang, Hans G. Othmer

**Affiliations:** 1Department of Mathematics, University of California, Riverside, CA 92507, USA; jgou@ucr.edu; 2School of Mathematics, University of Minnesota, Minneapolis, MN 55455, USA; zhan7594@umn.edu

**Keywords:** Hippo pathway, alpha-catenin, mechanical effects

## Abstract

**Simple Summary:**

Control of tissue growth is an important component of cancer treatment, but understanding how to do this effectively is still in the early stages. Tissue growth typically involves both complex interacting signal transduction networks as well as mechanical interactions within and between cells, and how they interact to control growth is an open problem. Herein we begin the process of how to understand the interactions by focusing on tissue growth in the fruit fly *Drosophila melanogaster*. A model of the Hippo pathway, which is one of the primary growth-control pathways, is integrated with a description of the mechanical behavior of cells in wing-disc tissue to predict how mechanics and signaling interact, and a number of significant insights and predictions have emerged from the analysis of the model.

**Abstract:**

*Drosophila melanogaster* has emerged as an ideal system for studying the networks that control tissue development and homeostasis and, given the similarity of the pathways involved, controlled and uncontrolled growth in mammalian systems. The signaling pathways used in patterning the *Drosophila* wing disc are well known and result in the emergence of interaction of these pathways with the Hippo signaling pathway, which plays a central role in controlling cell proliferation and apoptosis. Mechanical effects are another major factor in the control of growth, but far less is known about how they exert their control. Herein, we develop a mathematical model that integrates the mechanical interactions between cells, which occur via adherens and tight junctions, with the intracellular actin network and the Hippo pathway so as to better understand cell-autonomous and non-autonomous control of growth in response to mechanical forces.

## 1. Introduction

A long-standing problem in biology is to understand how the size and shape of multicellular organisms are controlled by the integration of signal transduction pathways that respond to organism-level growth control signals, local extra- and intracellular biochemical signals, and mechanical forces. In particular, understanding cell growth control in a given context is essential for understanding cancer and how particular types may be treated. For example, in the wing disc, ecdysone functions to regulate the disc size via the mTor pathway [1], and mutations or hyperactivation of signal transduction pathways such as the PI3K/Akt/mTOR pathway, the Hippo pathway, and the Ras/MAPK pathway are frequently involved in cancer. A difficulty in identifying type-specific treatments lies in the fact that signal transduction pathways such as these are often highly integrated with positive or negative feedback steps, which often provides a degree of signal transduction redundancy that makes discovering effective treatments difficult. For example, there is evidence that Akt is negatively regulated by Hippo signaling [2] (see Figure 1) and that inactivating Akt prevents entry of the co-transcriptional factor Yorkie into the nucleus [3]. Furthermore, the PI3K and MAPK pathways are strongly interconnected through a number of positive and negative feedback loops, and as a result, targeted inhibition of mTORC1 by treatment with rapamycin can cause MAPK reactivation and can lead to resistance to single mTORC1 inhibition. Consequently, combinatorial targeting of mTOR plus MAPK may induce a better response to therapies [4,5].

Further complicating the establishment of sufficient understanding to develop treatments is the more recently discovered fact that mechanical effects in the form of cell–cell and cell–ECM forces can also play an important role in controlling cell division and tissue size. Recent studies have shown that the nuclear translocation of yes-associated protein (Yap), a transcriptional co-activator which functions as a downstream effector of the Hippo pathway in mammals, is directly controlled by mechanical forces exerted through focal adhesions to the nucleus [6], which establishes a direct connection between substrate stiffness and cell growth. Furthermore, in epithelial tissues, strong cell–cell connections in the form of adherens junctions govern force transmission between cells and are essential in regulating contact-induced growth arrest and tissue integrity and homeostasis [7]. The primary components of the adherens junctions are the cadherin family proteins, and during tumorigenesis, one of the most important processes is the downregulation of E-cadherin, which leads to tumor cell migration and invasion [8]. Knowing the regulatory roles of the mechanical effects on the signaling pathways and the downstream gene expressions will better assist us both in understanding how cancer develops and in designing appropriate treatment strategies. Among the signaling pathways that are affected by cadherin cell–cell adhesion and contribute to tumor progression, the Hippo pathway is perhaps the best understood, due in large part to the high degree of homology between the Hippo pathway in *Drosophila*, particularly in the wing disc, and the Hippo pathway in mammalian cells (compared in Figure 2). A mathematical model of the Hippo pathway in the wing disc of *Drosophila* that can explain many of the observed results in mutant transplant experiments was developed earlier [9], and here, we extend this model and integrate it with a model for cell–cell interactions via E-cadherins and intracellular interactions via the actin network.

**Figure 1 cancers-15-04840-f001:**
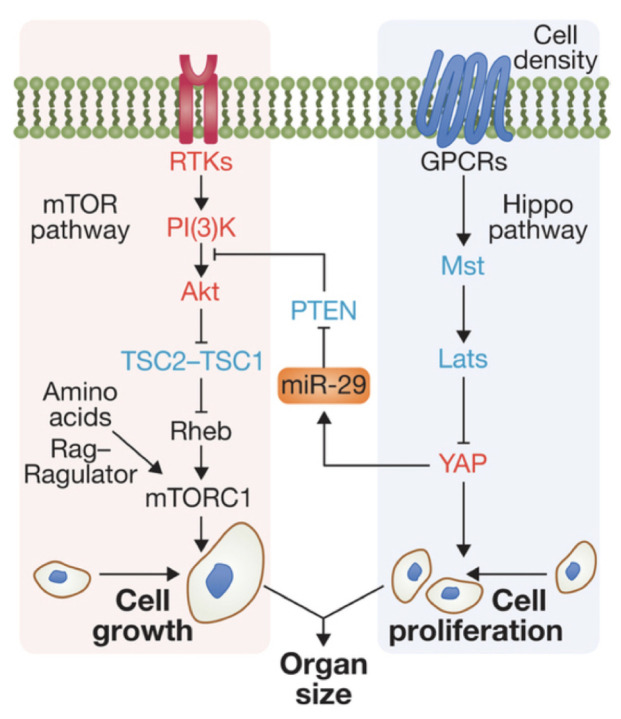
Interaction of the mTor and Hippo pathways in a mammalian cell. Modified from [10].

**Figure 2 cancers-15-04840-f002:**
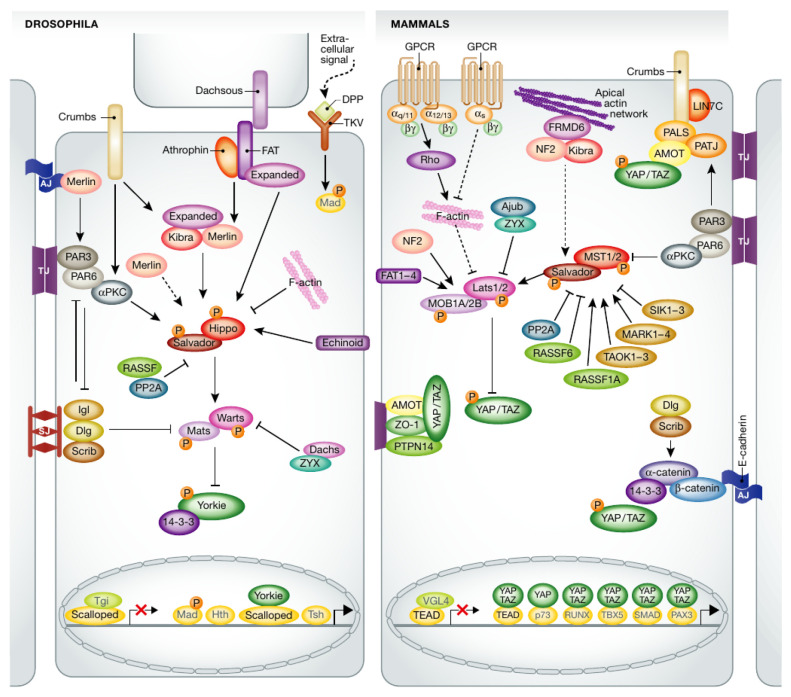
Comparison of the Hippo pathway in *Drosophila* and in mammals. From [11] with permission.

Our focus is on the development and analysis of a model of the *Drosophila* wing disc that integrates components of the known biochemical pathways and mechanical effects at the level of growth control so as to determine how normal control functions under various conditions and predict the consequences of dysfunction in pathways. A first step toward understanding the balances between biochemical pathways controlling Hippo activity was successful [9], and a major objective of this paper is to determine whether the current understanding of the biochemical and mechanical effects can explain effects such as disc over- or undergrowth following various interventions.

Several earlier models on integrating mechanics and signaling have been proposed [12,13,14,15]. In [12], the nonuniform growth in a layer of tissue was modeled, and it was proposed that feedback from mechanical stress provides a mechanism for stabilizing uniform growth. In [13], the authors coupled growth factor-induced growth in the disc and stretching-induced proliferation in the periphery of the disc and provided an explanation of the growth cessation in the *Drosophila* wing disc. Elsewhere, the authors modeled a regulatory network in the disc and coupled interactions between mechanical forces and certain growth factors [15]. A growth cessation mechanism depending on the absolute compression level in the center of the disc and the compression gradient across the disc was proposed. In the model described below, we will include the detailed cell–cell contact through adherens junctions and critical biochemical components in the Hippo pathway. Since disc growth involves the types of interactions found in many other epithelial tissue systems, we anticipate that detailed modeling of the disc will lead to general insights into development in mammalian systems and, in particular, insights into cancer development in epithelial tissues.

## 2. The Biochemical Regulation of the Hippo Pathway

The Hippo pathway is a highly conserved core kinase cascade that is regulated by many upstream factors and in turn regulates transcription factors involved in cell growth, proliferation, and apoptosis. The key effector in the Hippo pathway is Yorkie (Yki), a co-transcription factor whose nuclear localization is controlled by the kinase Warts (Wts). When Yki is phosphorylated by Warts, it cannot enter the nucleus. In the nucleus, Yki acts by binding to transcription factors such as Scalloped (Sd) to activate the expression of *cyclin E, myc, DIAP1*, and *bantam*, which control cell proliferation [16], and it also controls the expression of genes upstream in the Hippo pathway, such as Warts, Expanded, Fat, Ds, Merlin, Kibra, and four-jointed (Fj) (compared in Figure 3). Since Expanded (Ex), Merlin (Mer), Kibra, Fj, and Wts are involved in the regulation of Yki via the Hippo signaling, this establishes a negative feedback loop to regulate its own activity [17,18,19].

Two atypical cadherins, Fat (Ft) and Dachsous (Ds), are key upstream components in the Hippo pathway. Both Ft and Ds are transmembrane molecules that can be localized to the apical cell membrane. The intracellular domains (ICDs) of each can independently mediate signaling through the Hippo pathway within a cell, while Ft and Ds on adjacent cell membranes can bind together to form heterodimers which mediate cell–cell interaction and convey information between neighboring cells.

It was found that much of the Hippo signaling transmitted by Fat depends on the ability of its ICD to alter the subapical localization and activity of Dachs, an unconventional myosin. Dlish/Vamana, an SH3 domain-containing protein, binds with Fat and the Dachs regulator Approximated (App). When palmitoylated by App, Dlish helps localize Dachs to the subapical membrane, where Dachs can promote Warts degradation. Fat reduces the activity of App through direct binding and thus inhibits the subapical localization of Dachs [20]. Localized Dachs can bind the kinase Warts (Wts) and promote its degradation [21]. While overexpression of Dachs increases wing size, loss of Dachs completely suppresses the ability of a Fat mutant to induce overgrowth or target gene overexpression [22,23]. As Fat-Ds signaling also regulates the planar cell polarity (PCP), experiments suggest that while the polarization of Dachs controlled by Fat and Ds is essential for cell polarization, the amount of Dachs localized on the membrane controls cell growth [20].

The ICDs of Ds regulate growth through physical interaction with Riquiqui (Riq), an amino acid protein that contains five WD40 repeats that mediate protein–protein interaction, and Riq binds with Minibrain (Mnb), a DYRK family kinase. Ds is required for localization of Riq at the apical junction of cells in wing discs, and localized Riq does not affect Wts directly but rather potentiates the ability of Mnb to inhibit Wts through phosphorylation, thereby increasing Yki activity. Depletion of Riq or Mnb reduces Yki activity and results in smaller discs, while overexpression of Riq or Mnb increases the wing size [24]. Recent studies also suggest that Dlish/Vamana associates with the ICDs of Ds and Dachs, which provides another link from Ds to the regulation of Hippo signaling [25].

Interaction between Ds and Fat is a component of cell–cell interaction that is regulated by another protein, Four-jointed (Fj), which itself is one of the target genes of the Hippo pathway. Fj functions as a kinase that phosphorylates potential extracellular cadherin domains of Fat and Ds in the Golgi [26], thereby enhancing binding of Fat to Ds and inhibiting the binding of Ds to Fat [27]. However, the observed weaker phenotype of *fj* mutants compared with *ds* mutants, as well the high levels of Fat and Ds aggregation in the absence of exogenous Fj, indicates that Fat and Ds binding could occur even without Fj [27,28]. It was also found that either loss of Fj or overexpression of Fj across the disc causes modest reductions in wing size [29,30].

In addition to Fat-Ds signaling, another group of cell cortex-localized proteins, including Ex, Mer, and Kibra, also plays an important role in the regulation of the Hippo pathway through activation of Wts [31,32,33]. The apical-localized transmembrane protein Crumb, which is crucial in establishing and maintaining epithelial apical-basal polarity, directly binds to Ex through its FBM domain, and this localizes Ex to the apical membrane in epithelial cells, where Ex acts like a Wts activator [34,35]. In vivo, most Wts co-localizes with its inhibitor, the LIM protein Ajuba (Jub), at adherens junctions when Hippo signaling is inactive. When activated, Wts shifts from Jub co-localization to Ex association [36]. Ex can also directly bind with Yki to sequester Yki to the membrane and prevent its nuclear activity [33,37]. This effect was found to be independent of the Wts-mediated phosphorylation of Yki [37].

Merlin, a member of the ERM family proteins, and Kibra, another tumor suppressor, were found to activate Hippo signaling at the medial apical cortex in parallel with Ex. Together, they recruit the adapter protein Salvador (Sav) and Hpo, the *Drosophila* ortholog of the mammalian MST1 and MST2 kinases, to promote the activation of the Hippo pathway through phosphorylation and activation of Wts [32,38,39]. Another study also suggested that Mer could directly bind Wts at the plasma membrane, which promotes the phosphorylation of Wts by an Hpo-Sav complex. Although the details are still unclear, this Mer-Wts binding was shown to be promoted with the disruption of the actin cytoskeleton, which provides another link between the mechanical cues and growth [40]. As Crumbs forms complexes with Ex, Mer, and Kibra at apical junctions, which sequesters Kibra away from the medial apical cortex, it was suggested that Crumbs has a dual effect on Hippo signaling: it promotes the pathway activity through Ex recruitment and represses it by inhibiting the activity of Kibra [38].

The interactions between Ex and Fat in the Hippo pathway regulation remain poorly defined. A reduced apical Ex level was observed in a Fat mutant in spite of the increased *ex* transcription caused by Yki activity, and Fat is thought to be required for the normal localization and stabilization of Ex at adherens junctions [18,41,42]. Recent studies showed that Dlish/Vamana binds with Fat and Ex, and in vivo, it reduces the subapical accumulation of Ex. Loss of Dlish inhibits the destabilization of Ex caused by loss of Fat [20,43]. Direct binding between Ex and a highly conserved region of Fat ICDs, in parallel to Fat-Ds binding, was also found to recruit Ex to the apical junction and stabilize it [44]. However, another school of thought is that Fat and Ex function in parallel and have additive effects on imaginal disc growth and development. Although a Fat mutant induces partial loss of Ex protein from the membrane, mutation of Fat still promotes growth with overexpressed and accumulated membrane-localized Ex [45].

There are multiple time scales in the processes described above, ranging from extremely fast binding reactions to slower gene expression, and in the following model construction, we assume that gene expression is much slower compared with direct protein-protein binding or relocalization. This implies that the feedback regulation through gene expression controlled by Yki activity has a minor effect on the timescale under consideration, and thus it is not included in the model described below.

### Model Description


The complexity of the interactions described above precludes construction of a model that includes all of them, and to include only the essential steps in the regulation of the Hippo pathway, we developed a model that was modified from an earlier one described in [9]. The previous model focused on the collaboration of upstream regulators Fat and Ds and provided explanations to the seemingly contradictory experimental observations from Fat/Ds mutant experiments. The current model extends the earlier one by incorporating both regulatory mechanisms from Fat/Ds and Ex, and it is built around Fat, Ds, Fj, Dachs, Riq, Ex, Wts, and Yki. As the regulatory mechanism from Mer/Kibra was found to parallel that of Ex, their inputs are assumed to be constant for the problem of interest. Similarly, the binding between Crumb and Ex is not modeled explicitly.

The signaling network we considered is shown in Figure 3, and a detailed description of the model and experimental justification of the assumptions can be found in Section A.1. As the known regulation of the Wts inhibitor Jub is mainly from biomechanical signals, we first study the rest of the regulatory network for biochemical regulations and consider the integration in later sections. A brief summary of the major assumptions used in the model is given below:1.Fat and Ds bind together on adjacent membranes, where the cytosolic Fj acts on Fat to promote the binding and on Ds to inhibit it.2.Most protein–protein interactions are described by reversible binding and dissociation steps, except for those between Fat and Dachs, Ds and Riq, and Fj and Fat/Ds and the inhibition of Ex on the active Yki level. For instance, the inhibition by Fat of membrane localization of Dachs is modeled by a reduced binding rate and described by a decreasing Hill function of total membrane-bound Fat. The effect of Ds on facilitating the junctional localization of Riq is modeled in a similar way but with an increasing Hill function.3.The activation of Wts by Ex is modeled through direct, reversible binding of the two molecules. The level of the Ex-Wts complex is assumed to be proportional to the active Wts that is involved in Yki phosphorylation. The inhibitory effect of Ex on nuclear Yki activity is modeled as a separate inhibitory term in the governing equation for Yki.4.As the interactions between Fat and Ex are still poorly defined, we test the case where Fat stabilizes the junctional localization of Ex and compare it with the case where Fat and Ex act in parallel.5.As stated above, the feedback loops through the genetic regulations of Yki on Ex, Wts, and Fj were not incorporated in the current model.

The governing equations for all species can be found in Section A.1, and a representative equation that governs the Wts dynamics is shown below:$$\begin{array}{cc} \frac{d{\left[W\right]}_{c}^{i}}{dt}= & -k_{WA}^{+}{\left[W\right]}_{c}^{i}\sum_{j=1}^{N_{m}}{\left[Am\right]}_{m,j}^{i}+k_{WA}^{-}\sum_{j=1}^{N_{m}}{\left[\bar{WA}\right]}_{m,j}^{i}-k_{WR}^{+}{\left[W\right]}_{c}^{i}\sum_{j=1}^{N_{m}}{\left[R\right]}_{m,j}^{i}\\ & +k_{WR}^{-}\sum_{j=1}^{N_{m}}{\left[\bar{WR}\right]}_{m,j}^{i}-k_{EW}^{+}{\left[W\right]}_{c}^{i}\sum_{j=1}^{N_{m}}{\left[Ex\right]}_{m,j}^{i}+k_{EW}^{-}\sum_{j=1}^{N_{m}}{\left[\bar{EW}\right]}_{m,j}^{i}\;, \end{array}$$

Here $k_{WA}^{\pm }$, $k_{WR}^{\pm }$, and $k_{EW}^{\pm }$ are the binding/unbinding rates between Wts and Dachs, Riq, and Ex, respectively, and $N_{m}$ represents the number of separate membrane regions in the model. The mechanical regulation of Wts through Jub is not incorporated in the current equation. In accord with the previous model in [9], the current model predicts the inhibitory effect of Fat on the membrane localization of Dachs and the enhancement of membrane localization of Riq in Figure 4A.

Membrane-localized Dachs binds with Wts and promotes its degradation, while membrane Riq potentiates the Mnb-dependent Wts phosphorylation and reduces its activity. Together, they regulate the downstream level of active Wts, and in turn regulate the Yki activity. In addition, as the current model also includes the dynamics of Ex, we also tested how the Fat and Ex interaction affects the downstream level of Yki. In Figure 4B, the Yki level is shown as a function of the Fat production with the stabilization of Ex by Fat. When Fat and Ex function in parallel, Fat can only affect the Yki level through Dachs and Riq, where one is inhibitory and one has a promotional effect. When Fat also acts on membrane Ex, which promotes the activation of Wts, the inhibitory effect of Fat on Yki is strengthened, as shown in Figure 4B. In this case, further complications are added to the regulatory effect of Fat on the Hippo pathway. Furthermore, in the current model, we did not consider the effect of the Crb complex and the impact of Ex on the Hippo pathway through the interactions with Crb, and therefore Ex plays a solely inhibitory role on Yki.

The biochemical model presented here is a simplified version of the upstream regulatory network of the Hippo pathway. As noted earlier, the interactions between Ex and the Crb complex are neglected in the current model, as are their interactions with Hippo/Salvador, which phosphorylates and regulates the activity of Wts. The current model can be further extended by including the regulatory effect of these components and analyzing their impact on the downstream effector. As the Hippo pathway is highly conserved in *Drosophila* and mammals, a similar model framework can be used to study the regulatory mechanisms of the Hippo pathway in mammalian cells, and this will further help with the development of therapeutic strategies for cancer treatment.

## 3. Force Transmission within and between Cells

In an epithelial tissue such as the wing disc, tissue-level mechanical interactions are mediated via adherens junctions (AJs) and tight junctions (TJs), utilizing the former to transmit mechanical stress between cells and the latter to allow intercellular exchange of water, ions, and macromolecules and to block mixing of apical and basolateral lipids [46]. These contacts are critical for the establishment of apical-basal polarity, cell fate, and differentiation. In addition, as mentioned earlier, the atypical cadherins Fat and Dachsous form heterophilic bonds of the AJs apically and are involved in planar cell polarity [47]. Cell-level forces that arise in the cytoskeleton (CSK)—which is comprised of actin filaments, actomyosin, and other filament-based structures—are transmitted to the AJs via the cortical actomyosin network. This circumferential cortical bundle produces forces both normal and tangential to the membrane that are transmitted through the AJs, and these cell-to-cell connections in turn mediate tissue-level forces [48,49]. Much of the actomyosin in the wing disc is found in the cortical bundle—also called an actin belt in disc cells—which is located just below the apical membrane, and the apical actomyosin network [50]. The remainder resides in the CSK and in the actomyosin bound to lower portions of the cell membrane (see Figure 5). During growth, many regulators of actomyosin are upregulated, and the process of cell division involves precise regulation of actomyosin contractility [51]. Actomyosin levels are spatially variable throughout the disc, being lower in the center and elevated in the periphery and in cells along compartment boundaries [52,53,54]. As will be seen in the following, the role of actomyosin in exerting tensile force on AJs is crucial in the activation of many biochemical pathways, including the Hippo pathway [55].

The AJs provide an interface between the CSK and adjacent cells via E-cadherins (E-cads) [48,56] (compared in Figure 6). E-cads on opposing membranes form homophilic bonds with their extracellular domains, and these bonds have catch-slip properties, which means that their stability increases under tension. Adjacent E-cads on the same cell can dimerize via *cis* bonds by “swapping” their terminal regions, and the connection can be terminated by undoing this process [57]. Both tangential and normal forces acting on the AJs are transmitted to the contiguous cell, and these cell-to-cell connections in turn mediate tissue-level contractile forces.

The intracellular domains of the E-cads act as mechanotransducers that employ adapter proteins to connect them to the cortex and the CSK. The adapters include p120-catenin, α- and β-catenin (Armadillo (Arm) in *Drosophila*), and Vinculin (Vinc), which link the E-cads to the actin network (compared in Figure 6), and p120-catenin binds with E-cadherin at the inner surface of the membrane and interacts with myosin VI and RhoGTPase [59]. Arm binds directly to E-cadherin [60], and this occurs prior to E-cadherin reaching the membrane [59]. At the AJs, Arm forms part of the mechanotransduction link via its binding to both E-cadherin and α-catenin [49,61].

The tangential forces arise from actomyosin filaments in the actin belt aligned parallel to the membrane, while the normal forces arise from the CSK and from branched networks produced by Arp2/3 in the cortex [62,63]. The latter can generate protrusive forces that are important for maintaining stable cell-cell contacts at the AJs. Although the cortical belt to which the AJs are attached appears as a relatively homogeneous band of an actomyosin network along the cell junctions, there is additional internal organization. At a tricellular junction (TCJ) where three cells meet, specialized proteins are involved in cell division and other processes [64]. In particular, the protein Mud is involved in division orientation control [65], and loss of Mud randomizes division orientation in the wing disc, thereby inhibiting oriented division [66]. A distinct type of puncta forms adjacent to Jub/Vinculin puncta. Zyxin is recruited to junctions under tension, and its primary function is to repair actin in conjunction with Ena/Vasp [67,68,69]. In addition, Hippo pathway proteins have been found to be concentrated near the TCJs, and thus TCJs may serve as important signaling centers.

The response of the AJs to applied forces is anisotropic [70]; those in the direction tangential to the membrane tend to disrupt E-cad binding, whereas those in the normal direction can strengthen cell–cell adhesion via mechanical feedback. The strengthening of junctions under normal force and weakening under tangential force is a mechanism that serves to maintain epithelial integrity while allowing enough plasticity for division and cell rearrangements.

The actin nucleating and branching protein complex Arp2/3 is also found at AJs. It is believed to be necessary for AJ stability and formation by creating protrusive actin networks that push the AJ complexes on opposite cells together [62,71]. It was also found that the apical Crumb/Ex/Kibra/Mer complex responds to mechanical strains. When the apical area of the *Drosophila* follicular epithelium was stretched, reduction of the concentration of these complexes was observed, which in turn activated Yki activity [72]. But because the dynamics of Crumb, Kibra, and Mer, as well as their interactions with Ex are not considered in the current model, we assume constant inputs from these components in response to mechanical stretches and study the mechano-effect of adherens junctions separately.

### 3.1. The Structure of α-Catenin

The most thoroughly-studied mechanosensitive protein in an AJ is α-catenin [73,74,75]. It can bind both F-actin and β-catenin, and under tension it can undergo an “unfurling” change in conformation which exposes cryptic binding sites for Vinculin, actin, and Jub. As shown in Figure 7, α-catenin contains three domains:An N domain which can bind β-catenin and participate in the homo-dimerization of non-junctional α-catenin;An M domain (central modulatory domain) which contains the Vinculin binding site (VBS);An ABD domain, namely the C-terminal actin-binding domain.

A schematic diagram of these sites and the proteins that bind to them is shown in Figure 7. The M and ABD domains can undergo force-dependent conformational changes (the unfurling mentioned earlier) that can lead to force-dependent binding of various proteins. Several AJ-associated proteins can bind with α-catenin under different levels of tension [74], including the F-actin binding protein Vinculin [63] and the upstream Hippo pathway regulator Jub [76].

In *Drosophila*, binding of Jub to the N-terminal domain of α-catenin is tension-dependent [75,76]. The recruitment of Jub is negatively regulated by the M1 domain of α-catenin, but the binding rate does not depend on the presence of Vinculin [74]. Once recruited, junctional Jub sequesters Wts [77], preventing it from binding with the Wts-activating proteins Ex [36] and Crb [78] located apically to the AJs in the subapical region of disc cells. The effect of Jub on wing disc growth is known to be dynamic and asymmetric with regard to elevated or diminshed tension; younger discs are more sensitive to Jub underexpression, whereas older discs are more sensitive to Jub overexpression [52], and it is believed that most Wts is bound to Jub in late third-instar discs [36]. Interestingly, Jub also appears to be an essential component for AJ stability, possibly via a Rac-mediated mechanism that enhances actin association with E-cadherin complexes [79]. Other results show that Jub also interacts directly with F-actin [74,80]. This system exhibits multiple examples of feedback, as Jub mutants also cause Hippo-independent changes in myosin activity and may be involved in a negative feedback loop that downregulates tension via the activation of the cytohesin Step [81]. Finally, Jub may play a parallel role in the recruitment of Yki to the cell membrane, where it can activate myosin [82].

Although Vinculin does not affect the binding rate between Jub and the N domain of α-catenin, it does inhibit the folding process of the M1 domain of α-catenin. In addition, Vinculin can also exhibit a force-dependent enhancement of α-catenin binding to F-actin in that association of Vinculin with the ternary complex of E-cadherin, β-catenin, and αE-catenin increases the bound lifetime of individual complexes on F-actin as a function of the number of load-bearing complexes bound but only when force is directed toward the pointed end of the filament [83].

A summary of the force-dependent binding of Jub to α-catenin is given below [74]:M1 limits Jub recruitment, and loss of M1 causes Jub hyper-recruitment to AJs, promoting tissue tension-independent overgrowth. Although M1 binds Vinculin, Vinculin is not responsible for this effect.M1 normally limits junctional localization of Jub, while M2 and M3 normally appear to enhance junctional localization of Jub.The N domain is essential for Jub recruitment to AJs, but it was also found that the α-catenin-mediated recuitment of Jub to AJs does not seem to have a linear relationship with growth regulation in the wing disc epithelium, as normal growth is compatible with normal or low concentrations of Jub at AJs.

A summary of the effect of tension sensing is shown in Figure 8, where there is a Jub-dependent mechanism ① and an unknown Jub-independent mechanism for growth regulation ②.

The authors show that stability of the cadherin-catenin complex (CCC) is controlled by the M2 domain and the α-catenin actin-binding domain (ABD) and that lower CCC levels reduce Jub recruitment.

### 3.2. A Three-State Model for Binding and Unfurling of the AJs

Because there are different energy wells associated with the different states of α-catenin, the response to tension involves discontinuous jumps between these states under the force-driven transitions of α-catenin. An early model to describe cell–cell adhesion was developed by Bell [84], who used an exponential function to describe the lifetime τ of molecular bonds. This takes the form
(1)$$\tau =\tau_{0}e^{\left(E_{0}-\gamma F\right)/k_{B}T},$$
where $\tau_{0}$ is the reciprocal of a natural frequency, $E_{0}$ is the bond energy, *F* is the applied force per bond, and γ is a parameter that must be determined empirically to account for the structure of the material. This model has been used to describe protein folding and unfolding under force (reviewed in [85]) and leads to the following expression for the rate constant of protein folding and unfolding:$$k\left(F\right)=k_{m}k_{0}e^{Fx/k_{B}T},$$

Here k(f) is the rate constant under a force *f*, $k_{m}$ includes the contributions of the components of the system to the observed rates, $k_{0}$ is the intrinsic rate constant in the absence of force, *F* is the applied force, and *x* is the distance to the transition state. If the forces for transitions are known, then one can approximate the value of $k_{*}^{0}\equiv k_{m}k_{0}$ [86] (see Section A.2).

We can define three distinct states of α-catenin: (1) only the N region is exposed, and all M regions are folded; (2) the N and M1 regions are exposed; and (3) the N and all the M regions are unfolded. Here, we assume that the M2 and M3 regions fold and unfold together, since current experiments have not identified a clear transition state between these two states. Based on this, we can classify the E-cad-β-cat-α-cat complex into three different states: $CCC_{1}$, $CCC_{2}$, and $CCC_{3}$. This leads to the system
(2)$$CCC_{1}\overset{k_{u1}\left(f\right)}{\underset{k_{f1}\left(f\right)}{⇌}}CCC_{2}\overset{k_{u2}\left(f\right)}{\underset{k_{f2}\left(f\right)}{⇌}}CCC_{3},$$
where
(3)$$\begin{array}{cc} k_{fi}\left(f\right) & =k_{fi}^{0}e^{-fx_{fi}/k_{B}T}\\ k_{ui}\left(f\right) & =k_{ui}^{0}e^{fx_{ui}/k_{B}T},\;i=1,2. \end{array}$$
and $CCC_{1}+CCC_{2}+CCC_{3}=1$. Hereafter, the states are labeled by dropping the leading CC. Further details on the development of the equations can be found in Section A.2.

In the following computations, we hold all components in Figure 9 but α-catenin constant and vary the forces to determine the steady state levels of the three states. At zero force, $C_{1}$ was open to Jub binding, but as the force increased, α-catenin started unfolding from its initial (fully folded) state (Figure 10A), and exposure of M1 led to inhibition of Jub binding. A crossover between the decreasing $C_{1}$ state and the increasing $C_{2}$ state occurred at a force of ≈4.5 pN. A similar crossover from M1 into the M2 and M3 unfolded states occurred at ≈10 pN, which accords with the experimental result [87]. Furthermore, the time for this simple module to reach a steady state was around 2∼4 s, which also accords with the experimental result [87]. Figure 10B shows how the bound and unbound Jub depended on the force. The level of bound Jub was largest when the force was large and at its second largest when the force was small. In the range from 5 pN to 9 pN, where the M1 region was unfolding, the amount of bound Jub was lowest because M1 inhibited Jub binding.This quantifies more precisely the known result that the amount of bound Jub varies with the level of force [74,88].

We show the crossover points for the state transitions as a function of Vinculin in Figure 11, where one sees that Vinculin enhanced the $C_{1}$-to-$C_{2}$ transition by reducing the force required. Although Vinculin does not affect the binding between Jub and α-catenin, it can inhibit the folding from $C_{2}$ to $C_{1}$, which agrees with our expectation.

### 3.3. A Mechanical Model for Tissue-Level Interactions

Next, we introduce a model for a tissue that incorporates both the mechanical interactions within cells that occur via the actomyosin network at the apical membrane and the interactions between cells via the cortical belt that connects TCJs. Because the quantities involved in growth control via Yorkie are predominantly localized at or near the apical membrane [36,89], we focus on a two-dimensional section of a cell that includes the apical membrane and a small vertical component of the lateral membrane where the tight junctions and AJs are found. We furthermore assume that the cytosolic fluid is quiescent, and therefore the disc mechanics are determined by the mechanical properties of the actin belt and the cyctosolic component of the actomyosin network. It is known that on a longer time scale of days, the disc mechanics change with age, with younger third-instar wing discs held in a higher tension state, and that central compression occurs as the discs age due to rapid growth in the disc’s center [52,76]. The time scale used here is in the range of seconds to minutes (for example, Jub binds to α-catenin in seconds [90]), and the results indicate the short-term changes in the network that resulted from changes in various factors rather than the final state.

We studied a small network of cells—a central cell and six neighbors, as shown in Figure 12A—to understand how stress and imposed growth interact at the single-cell level. All cells were equal-sized hexagons initially, and we prescribed the elements in each cell as shown. The nodes received forces from the belt connections in the network and transverse forces from other nodes in their cell that reflected the actin network attached to the apical membrane. How the forces were computed is described in Section A.3.

A model that incorporates details of the actin network growth and its interaction with myosin and numerous control molecules such as capping proteins would be far too complex to implement here [91,92,93], and thus we describe the passive components of the network with a Kelvin–Voigt (KV) viscoelastic model. We combined this with an active element for the contractile actomyosin network as shown in Figure 12B, and we refer to the combined module as the KVA model. The form of the active component is given in Section A.3.

The KV model represents elastic interactions and viscous damping within and between cells, and it was chosen because it describes the stress response of *Drosophila* embryonic wing cells well [94], and because it is the simplest of the standard models in which the tissue relaxes to its original configuration when the stress is released. Cell-ECM interactions are not considered here, but their importance for tissue pattern formation has been investigated [95], and it was found that the stress-strain constitutive equations used for the ECM play an important role in that context.

Since tissue rheology has been studied extensively, there are good estimates of the properties of the cytosol [96], the cortex (including cortical tension) [97,98], and the effects of perturbations of actomyosin activity [99]. This level of detail in the model is necessary for several reasons. Vertex-based models have been widely used for modeling epithelial sheets [100,101,102], but they are inadequate here because they ignore active stress [103,104], since cell surface tensions—represented here by the cytosolic connections—are comparable to cell-cell interface tensions in embryonic *Drosophila* tissue [105], and because such models assume instantaneous relaxation of mechanical interactions, which may not hold in wing disc tissue in general [106].

To develop the governing equations, we defined two matrices to describe the connectivity in the tissue: one ($\Delta_{N}$) for the actin belt connections between nodes and another for the transverse connections that represent the actomyosin network underlying the apical membrane, which we call $\Delta_{T}$. Formally, these represent the Laplacian operator on the underlying graph of the network, and we introduced two matrices because different constants may appear in the node-to-node connections and the transverse connections. Under the earlier assumptions, the structure of the governing dynamical equations for the network represents a force balance given by the following momentum equation:(4)$$\begin{array}{cc} M\frac{d^{2}x}{dt^{2}} & =K_{N}\Delta_{N}\left(x-l\right)+K_{T}\Delta_{N}\left(x-l\right)+\Xi \frac{dx}{dt}+F_{A}\left(\frac{dl}{dt}\right)+F_{Imp}. \end{array}$$

Here, *M* is the diagonal matrix whose entries are the masses of the nodes, x is the vector of the spatial positions of the nodes in $R^{2}$, l is the rest length of the springs, x−l is the vector of spring extensions, $K_{N}$ ($K_{T}$) represents the diagonal matrices whose entries are the elasticity coefficients for the belt connections (transverse connections), and Ξ is the matrix of damping coefficents for the viscous response. When all nodes have the same properties, the matricies $M,K_{N},K_{T}$, and Ξ are replaced by constants. The term $F_{A}$ represents the active force, and $F_{Imp}$ represents an externally imposed force. Details for these and other terms in Equation (Equation 4) are given in Section A.3.

In addition to the mechanical dynamics, we allowed for growth of the nodal and transverse connections, and this led to the equation
(5)$$\begin{array}{cc} \frac{dl}{dt} & =F_{G} \end{array}$$
where $F_{G}$ represents the growth function. In reality, this can depend on numerous factors such as capping proteins, cofiln, and others that affect the growth of the actomyosin network [91,92], but we used the simplified form given in Section A.3.

As an illustration of the effect of abnormal growth of one cell under purely mechanical conditions—the biochemical network is turned off—we began with a configuration of equal hexagonal cells in which the active and passive forces were balanced. We then artificially increased the size of the central cell, which was intended to model a volume change of a cell that had taken up water due to osmotic changes, and computed the new stationary state when the system relaxed. The results are shown in Figure 13, where here and hereafter, forces applied at the exterior nodes were directed along a ray connecting the node and the center of the network. One can see that in each case, the swelling of the central cell changed the surrounding cells into irregular hexagons (i.e., hexagons of unequal sides). In Figure 13b, the contractile forces shrank the network. (We did not impose volume conservation to reflect the fact that cells can adjust their volume by controlling the water content via tension-dependent aquaporins.) Application of a stretching force of 800 pN per node in Figure 13c had a dramatic effect on the size, whereas a compressive force of 100 pN in Figure 13d had little effect.

## 4. The Interaction of Biochemical Signaling and Mechanics

In this section, we combine the biochemical and biomechanical models developed earlier, the result of which we call the biochemical signaling and mechanics (BSM), and we compare simulation results from the BSM with experimental observations for different conditions. We have seen that α-catenin plays a central role in transmitting signals between the mechanical and biochemical components, which in turn respond with direct or indirect feedback. We have focused on α-catenin, but it should be noted that the experimentally observed junctional complex includes other components such as E-cad and β-catenin, and the formation of the complex is regulated not only by force [48,58,107,108] but also by biochemical factors such as Vinculin [109,110,111]. Here, we ignore the formation of junctions and only study their role in regulating junctional Jub localization, which in turn regulates Wts and Yki. Using the BSM, we aim to understand the experimental observations related to regulation of the Hippo pathway by mechanical inputs and to obtain insight into how their interactions can control cell and tissue growth. Possible extensions of the current model are discussed in Section 5.

Based on the studies of α-catenin described in Section 3, we assume it has three possible states: $C_{1}$, $C_{2}$, and $C_{3}$, which correspond to the degree to which it is unfolded. In the simulations, we assumed that the total amount of α-catenin was maintained at a constant level, and in light of the fact that the expression level of E-cad in epithelial cells is approximately 26–46 molecules/μm${}^{2}$ [112], we assumed that the membrane-localized α-catenin was maintained at a similar level. In addition, as most of the E-cad was localized at AJs in a region which was around 1 μm below the apical membrane [113], and the averaged edge length of the cell edges in the *Drosophila* wing was around 1 μm, we assumed that the total amount of the different states of α-catenin was about 26–46 molecules per cell edge. In our model, the mass is carried by vertices as shown in Figure 12, and therefore the total number of α-catenins was set to 26–46 molecules per node. Furthermore, despite the fact that the level of membrane-localized α-catenin can change dramatically at different stages of cell development [113], we will see that the qualitative model predictions will not change with changes in α-catenin.

In the diagram shown in the left panel of Figure 14, we illustrate qualitatively the Yki activity level as a function of the tension forces acting on an α-catenin molecule. This shape can be understood by comparison with the level of bound Jub in Figure 10B, because when Jub is low, Wts will be high, and Yki will be low. The dip in Jub at intermediate forces stems from the fact that when M1 is open, it inhibits binding of Jub to the N-domain of α-catenin (see Section 3), and this inhibition is suppressed at sufficiently high tensions. Thus, the system can be in one of three states denoted by 1, 2, and 3, in which the relative level of Yki first decreases from stage 1 to 2 and then increases from stage 2 to 3. While this qualitative description is robust, details may change as parameters are changed. For example, the level of Vinculin controls the location of the transition from stage 1 to 2, as shown in Figure 11. Increasing Vinculin decreases the transition force needed for a transition from $C_{1}$ to $C_{2}$, which will shorten the 1 stage in Figure 14 (Left) as a result. Furthermore, cells may experience different cytoskeletal tensions and possess different levels of α-catenin in different developmental stages [110,113], and their states may progress along a non-monotonic path in the x direction of Figure 14. This will be elaborated upon later.

In addition to analyzing the BSM under different cytoskeletal forces, we also explored how the system responds to changes in α-catenin. For this purpose, we constructed a growth model for cells shown in the right panel of Figure 14 and assumed that the cell area grew following a Yki-dependent equation described in Section A.3. As an initial condition, we assumed that the volume of the center cell enlarged from the region enclosed by black dashed lines to that enclosed by red dashed lines due to, for example, changes in osmotic conditions. Tension and compression forces can also be applied to the boundary nodes, as in the pure mechanical example shown in Figure 13. In any case, one must now solve the system comprising Equations (Equation 4) and (Equation 5).

According to the three-state model of α-catenin described earlier, increasing a tension force from zero should first produce a decrease in Yki, as shown in Figure 14, followed by an increase at larger forces. However, experimental observations show that the Yki level increased with increasing cytoskeletal tension [76]. Based on the model, we suggest that this discrepancy between the molecular- and cell-level observations may be due to the magnitude of the imposed force. Indeed, when a single α-catenin is bound with E-cad and β-catenin, a force of around 5 pN is needed to unfold the M1 region (although when vinculin exists, this value may decrease) [87], which converts the α-catenin to state II. Given that the tension force generated by a single Myo-II molecule is around 3–5 pN [110], it is possible that in cell-level experiments, α-catenin unfolds the M1 region directly without passing through state I in the left panel of Figure 14. To test this hypothesis, we plotted the Yki activity as a function of the cytoskeletal tension and the level of membrane-localized α-catenin in Figure 15. When the α-catenin level was low (20–50 molecules/node, which was the normal range we set in our system), the Yki level increased in response to increasing force (right lower panel of Figure 15). However, if the α-catenin level increased to >50 molecules/node, then the transitions were those predicted by the molecular-level mechanism, as shown in the upper right panel of Figure 15 [74,75,88]. This is due to the fact that as the level of α-catenin increases with the same magnitude of tension applied to each node, the force experienced by individual α-catenin molecules decreases. Further insight can be obtained from a vertical slice in the heat map shown in Figure 15. When the magnitude of the applied tension was fixed at, for example, 600 pN/node, Yki would first increase and then decrease as α-catenin increased. Since α-catenin modulates the Hippo pathway through membrane localization of Jub, which is a negative regulator of Warts, an elevated level of α-catenin promotes Yki activity at intermediate levels. It was also observed that a loss of α-catenin can cause a decrease in the Yki level [74,76], which leads to tissue undergrowth. Beginning with α-catenin at 60–70 and the tension fixed at 300 pN/node, one sees in Figure 15 that Yki is essentially monotone decreasing as α-catenin decreases, which comports with the observations. It was observed that the E-cad level increased as the wing disc cells aged [113], and the localized α-catenin level may also increase. If the tension does not change proportionally the Yki activity will be inhibited by the increase of α-catenin and possibly lead to cell death.

We further investigated the effect of the α-catenin level on Yki activity with imposed compression and tension forces. For the results shown in Figure 16, we varied the amount of α-catenin in the center cell while keeping the α-catenin in the boundary cells constant. One can see there that by fixing the boundary tension, increasing α-catenin in the center cell can induce higher Yki activitiy there, which again agrees with the experimental observation described earlier. One can also see that at low α-catenin levels, Yki increased more slowly at the center under tension than compression, but the difference disappeared at sufficiently large α-catenin levels. Interestingly, increasing α-catenin in the center cell had a negative effect on the Yki level in the boundary cells under both tension and compression. We could not solve the governing system analytically, but this effect was caused by the mechanical feedback among cells in the tissue, which Yki would transduce into an effect on the growth. This observation provides a possible explanation for the observation of cell-autonomous decreases and non-cell-autonomous increases in Yki activity in tissues with E-cad or α-catenin knockdown [114].

Next, we investigated the effect of α-catenin on the discrepancies in the Yki level and cell growth in the center and boundary cells. As shown in Figure 17, with a higher α-catenin level, the difference in the Yki level and the growth between the center and boundary cells increased. The effect of the magnitude of the applied environmental force only had a significant effect when the α-catenin level was relatively low. To understand this result, we should first notice that a lower boundary tension will decrease the cytoskeletal tension, especially in this case, where the boundary tension varied from 800 pN to 100 pN. Furthermore, as shown in Figure 15, if α-catenin is in the normal range (from 20 to 50), then the system will generate an increasing Yki level as the cytoskeletal tension increases, but if the α-catenin amount is higher (>50), then the system may produce the same high Yki level at lower cytoskeleton tension as a system with high cytoskeletal tension.

We also analyzed the model under a similar setting with and without cell growth, as shown in Figure 18. After the same evolution time for each case, with cell growth, the Yki level was always lower, although slightly, compared with that without cell growth for both magnitudes of the applied environmental forces. We think this was caused by the unbalanced growth; the center cell had a higher growth rate than the boundary cell, which could also be reflected by the Yki level. This unbalanced growth will increase the compression between cells in the tissue and, as a result, inhibit the Yki activity compared with the no-growth cases. This observation can be interpreted as the feedback of the biochemical module on the mechanical changes. Since the growth of the center cell caused expansion of the cell area, it compressed the surrounding cells. In return, the Yki level in the center cell decreased as a responses to mechanical cues to avoid tissue overgrowth and overcompression. Such feedback has been widely explored both theoretically and experimentally [7,115,116,117,118,119], and our model provides a platform for further investigation of this mechanism. However, as shown in Figure 18, this effect was not all that significant in the model results, and this was caused by the relatively simple description of growth. The real growth process is more complicated, with dynamics of the cell size, actin, etc. such as that shown in [120]. A more comprehensive model with the incorporation of a detailed growth model could amplify this feedback effect and provide a better understanding of the details.

As discussed earlier, α-catenin plays a central role in bridging the mechanical and biochemical signals in growth control by regulating the membrane localization of Jub [76,80]. As shown in Figure 10, we used a single-molecule model to show that under higher cytoskeletal tension, Jub accumulated at the AJ, which accords with the experimental observations [76,121].

We could also analyze the difference in Yki level and growth between the center and boundary cells as functions of the α-catenin level when different Jub expressions were imposed (Figure 19). This setting could be used to mimic the case where a patch of Jub overexpressed or underexpressed cells surrounded by WT cells. As shown in Figure 19, by fixing the environmental force at 800 pN, we observed that the Yki level and growth effect both increased as the Jub production rate increased, which accords with the experimental observations that increased Jub recruitment to α-catenin is associated with increased Yki activity and wing growth [75].

## 5. Conclusions

### 5.1. Overview


Studies on vertebrate systems showed that signal transduction and growth control pathways are highly conserved across species, and thus much of what is learned about *Drosophila* applies in higher organisms. Tissue overgrowth in *Drosophila* is similar to tumor growth in mammalian systems, and a better understanding of how molecular signals and mechanics interact in development will shed light on how these signaling processes interact in the microenvironment of a tumor to affect its growth [122,123]. The ubiquity of the pathways across species and the balances of signals between them within an organ highlight the need for mathematical models to understand these complex systems.

Growth control in the *Drosophila* disc involves both disc-level and extrinsic control [124], and since Yki-dependent growth requires dTOR activity, the extrinsic control structure may depend on a hypothetical mechanism in which dTOR affects the state of nuclear Yki [125]. The dTOR and Hippo pathways may act in parallel in that Hippo assesses local growth suitability, and dTOR ensures that organismal factors such as nutrition are sufficient to support the increased growth in the early stage of the cell cycle. In [126], we suggested a description called the mechanical feedback model (MFM), in which mechanics and signaling are integrated for describing disc growth, but while the MFM incorporates part of the Hippo pathway, a description of growth also requires the integration of extrinsic growth controls transduced via the dTOR pathway [127].

Mechanical effects due to an influx or efflux of water may also be important. As we showed in the example in Section 3.3, enlargement of a cell can have a significant effect on the cells connected to it, and this example suggests that osmotic effects can have an important impact, as noted in several recent reviews [128,129,130]. In particular, recent research has shown that volume and shape control can play a significant role in cancer cell migration, and this involves a variety of control factors [131,132,133,134]. Ion channels and aquaporins are central components of cell volume regulation, and water flow driven by osmotic gradients generated by ion transport contributes to the driving force for cell migration. In a simple experimental example, it was shown that tumor cells confined in a narrow channel can translocate by establishing an asymmetric end-to-end distribution of Na+/H+ pumps and aquaporins, which creates a net inflow of water and ions at the cell leading edge and a net outflow of water and ions at the trailing edge [135]. Moreover, it has been reported that metastatic cancer cells have higher expression of these transport proteins than nonmetastatic cancer cells [136].

### 5.2. Current Results

Herein, we developed a model for tissue development in the *Drosophila* wing disc that integrates an earlier model of the Hippo pathway with a model of the tissue mechanics. In light of the complexity of the actomyosin network and the cell-cell connections that govern the mechanics, it was necessary to use the high-level Kelvin–Voigt description of the cortical and cytoskeletal dynamics and a simplified model of cell-cell connections. Nonetheless, a number of significant insights have emerged and a number of important results have been obtained despite the simplifications.

In Section 3, we developed a model for the unfolding of α-catenin under increasing force levels, and in particular, we showed how the transition between the $C_{1}$ and $C_{2}$ states depends on the level of Vinculin. This and other results reported herein involving the unfolding process provide a starting point for further investigations of the dynamical response of the cell-to-cell connection machinery based on E-cads, α-catenin, and other components. The universality of the components involved in cell–cell adhesion suggests that what is learned from *Drosophila* will be applicable to other systems and may shed light on alterations in the interactions that arise in cancer.

In Section 4, we performed numerical experiments that validate our model by comparison with experimental results and enabled us to make testable predictions. It has been widely reported that α-catenin is a critical component that connects the mechanical cues with the Hippo pathway, but the detailed mechanism is not well understood [75]. Recently, with the analysis of phenotypic defects resulting from the differential reduction of gene function [74,88], it was found that different binding regions of α-catenin have different effects on the binding between α-catenin and Jub. Furthermore, the folding and unfolding behavior of α-catenin was found to be force-dependent [87,137]. Based on these findings, we constructed a model that describes the conversion between different states of α-catenin and employed Bell’s model to describe the force-dependent transition rates. The simulation results were compared with experimental observations, which showed that the model successfully reproduced the force-regulated folding and unfolding of α-catenin.

Next, we integrated this model with the model for the Hippo pathway, and from this, were able to reproduce the effect of elevated α-catenin and the cytoskeletal tension on Yki activity. In addition, we can explain a critical discord between the molecule-level and cell-level observations: Although the molecule-level experiments predicted three distinct stages of force-dependent Yki activities, in the cell-level experiments, only the latter two were observed [76]. We suggest that this discrepancy may be due to the relatively high magnitude of forces exerted by Myo-II on single α-catenin molecules, which implies rare observation of the first stage. We can also speculate, based on this analysis, that in older cells, a higher E-cad level may reduce Yki activity, which could relate to cell death.

Finally, integration of the mechanical and biochemical model with a model for growth enabled us to reproduce many growth-related observations, including the role of α-catenin level-dependent tissue growth and the effect of Jub expression on growth. The results show that the integrated response of the mechanical, biochemical, and growth modules to variations in the α-catenin level across a tissue emphasized its role as a critical pivot in tissue growth regulation [7,115,116,117,118,119].

In summary, the interaction between the Hippo pathway and the mechanical forces can lead to control of growth in the *Drosophila* wing disc under various conditions, although a more detailed model is needed to investigate the mechanical effects further.

### 5.3. Future Directions

In light of the fact that this is the first attempt known to us to model this complex system, we have indicated a number of potential directions for future investigations.

As stated earlier, in the current model, we ignored the feedback loop in which Yki activity activates the expression of upstream regulators, such as Four-jointed, Ex, Mer, and Kibra [138], and a similar feedback loop exists in that of mammalian cells [139]. On a longer time scale, this feedback may play a significant role in controlling growth.A more detailed model of the actomyosin dynamics is needed to more accurately describe the tissue mechanics. While much is known about individual components, a cell-level model that describes the details of the actin belt dynamics and its interaction with the E-cad-α-catenin system is not at hand. For example, a more detailed model would incorporate spectrin, a contractile protein that forms a membrane-attached skeleton beneath the plasma membrane by crosslinking short F-actin and binding-integral membrane proteins [140]. Spectrin is required for the formation of epithelia, and unlike other regulators such as Crumbs and Merlin, it regulates Hippo signaling by modulating cortical actomyosin activity through non-muscle myosin II [141].A more detailed and dynamic model of the mechanics of cell–cell interactions is needed, including the binding of E-cadherin between cells and α-catenin binding and production or mobilization, is needed. This is particularly important for understanding relative movement between cells and T1 transitions.Because the copy numbers of signaling molecules and other key components are frequently low, stochastic effects should be considered with a view toward understanding if and how the network structure plays a role in adapting to noisy signals. Noise can affect the precision of gene expression in simple networks [142], and key components of the Dpp pathway, for example, are present at nanomolar concentrations [143] in the disc, yet the disc patterning and size are remarkably reproducible. Though it has been argued that even such low concentrations are sufficient to mitigate stochastic noise [144], the network structure may play a role in mitigation. How the functions of signal transduction networks and mechanical regulation are maintained in the presence of fluctuations is still a major question in cellular biology.The current model involves regulation of both the Hippo pathway and the mechanical pathway, and in both components many of the parameters used in the model are unknown or difficult to measure in experiments. Therefore we selected the parameters within biologically-meaningful ranges, but did not do a detailed sensitivity analysis. In the previous version of the biochemical model [9], of which the current biochemical model is a small variation, both local and global sensitivity analysis was performed to analyze the impact of the parameters in the results, and it was found that a small number (6–8) of parameters were very important, and the majority were much less important. However, the Hippo signaling network in both cases has a top-down structure and feedback loops are not considered, which may affect the results. For instance, it was found earlier that the variations of parameters close to the downstream output – the cytosolic Yki concentration – have a greater effect on the variations of the output than do the upstream parameters.A similar sensitivity issue applies to the mechanical component of the model, and when it has been done, the sensitivity analysis for the combined model has to be done. This has the potential to identify the key steps in the combined model and lead to a reduced model that can be used in tissue-level computations, but this is a major project in its own right, in part because how the parameters affect the qualitative responses, for instance, the Yki profile as a function of Fat production as shown in Figure 3, is difficult to analyze and required a method that differs from the traditional global and local sensitivity analysis.

## Figures and Tables

**Figure 3 cancers-15-04840-f003:**
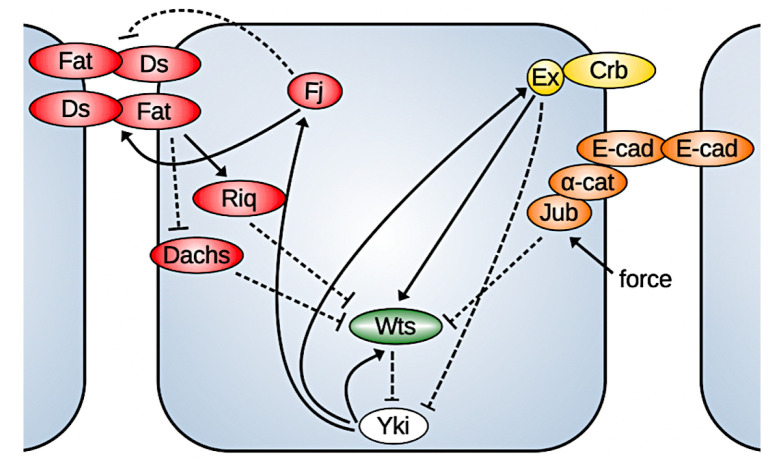
Basic components of the Hippo pathway described herein. Positive feedback steps occur on a slower time scale.

**Figure 4 cancers-15-04840-f004:**
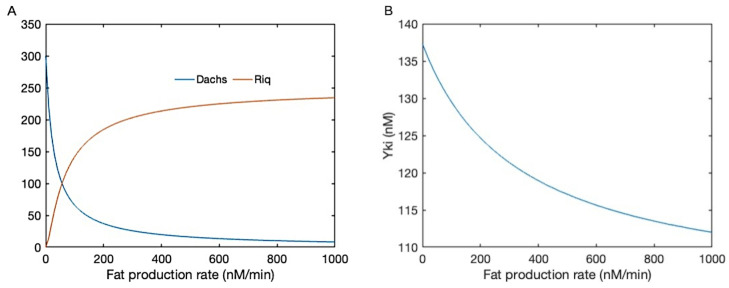
(**A**) The membrane-localized Dachs and Riq as a function of Fat production. (**B**) The active Yki level as a function of Fat production with Fat-dependent stabilization of membrane Ex.

**Figure 5 cancers-15-04840-f005:**
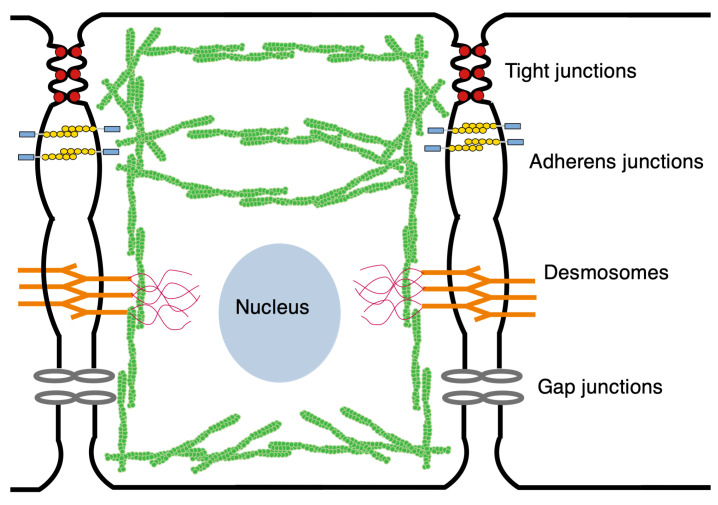
The mechanical structure of a wing disc cell. Green denotes the actin network.

**Figure 6 cancers-15-04840-f006:**
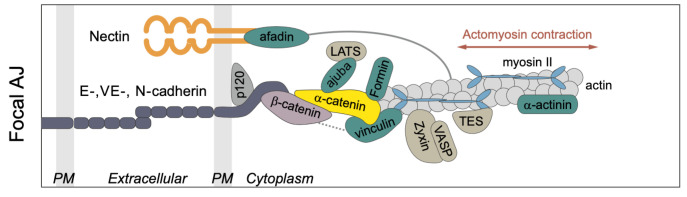
Components of the junctional complex. PM = plasma membrane. Taken from [58] with permission.

**Figure 7 cancers-15-04840-f007:**
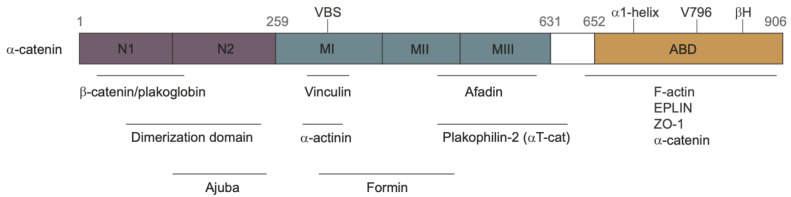
The domains of α-catenin. VBS = Vinculin binding site; ABD = actin binding domain. Taken from [58] with permission.

**Figure 8 cancers-15-04840-f008:**
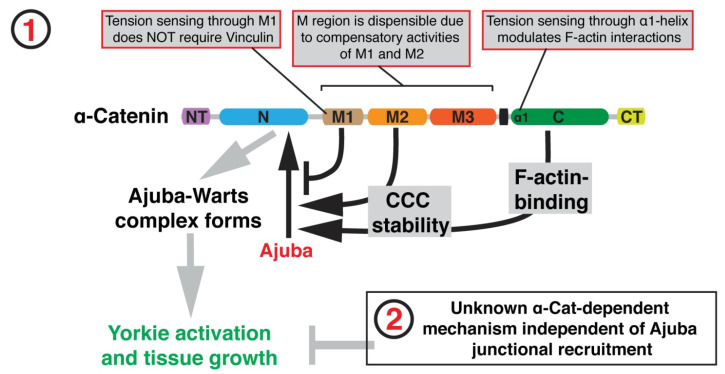
A model of how binding of Jub to α-catenin controls tissue growth in the *Drosophila* wing disc. Taken from [74] with permission.

**Figure 9 cancers-15-04840-f009:**
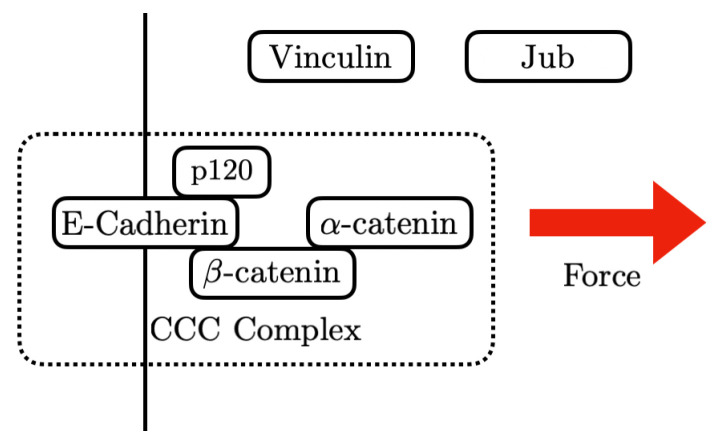
A simple model used to simulate the AJ under different forces.

**Figure 10 cancers-15-04840-f010:**
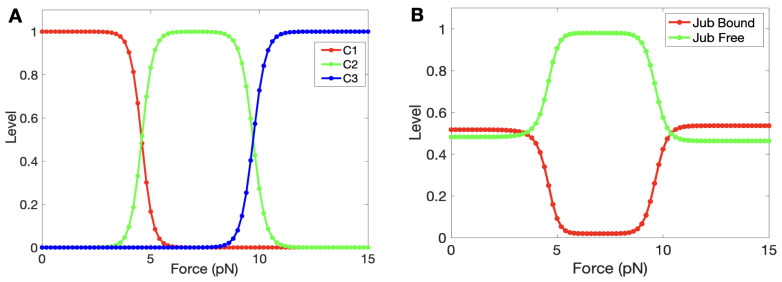
(**A**) The progression of the states as the force is increased. (**B**) The variation between bound and unbound Jub as a function of force.

**Figure 11 cancers-15-04840-f011:**
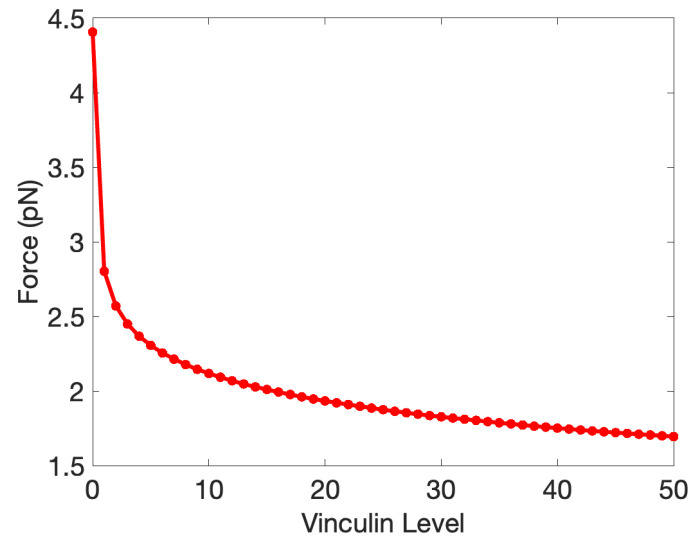
The force for the crossover from $C_{1}$ to $C_{2}$ as a function of Vinculin. The crossover from $C_{2}$ to $C_{3}$ is Vinculin-independent.

**Figure 12 cancers-15-04840-f012:**
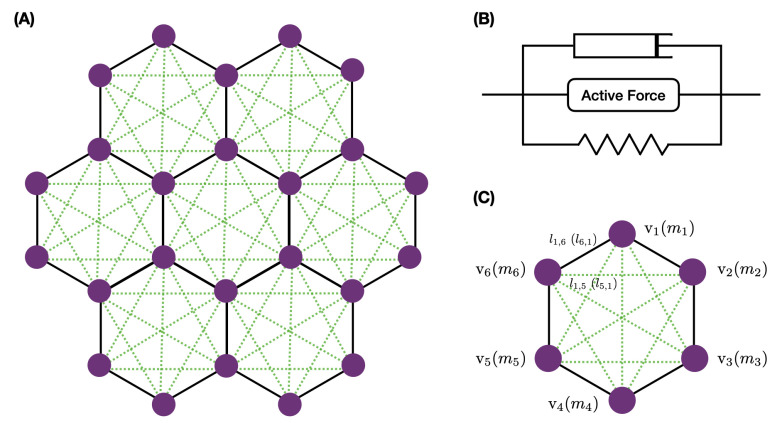
The basic network (**A**), the KVA model (**B**), and the details of the notation (**C**), where $v_{j}$ denotes a node and $m_{j}$ its mass.

**Figure 13 cancers-15-04840-f013:**
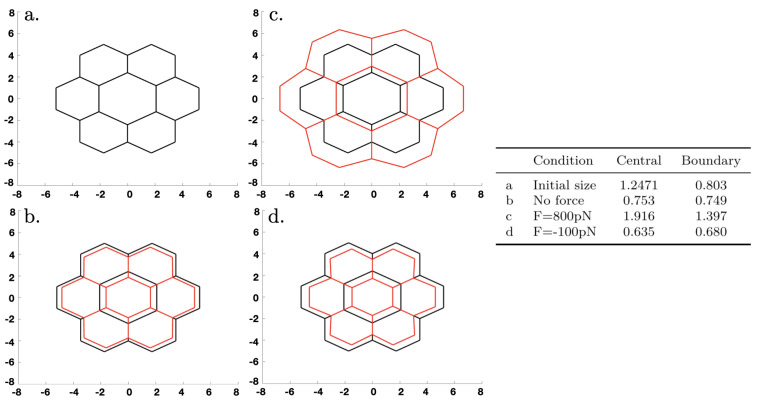
The dynamics under an initial increase in the size of the central cell. Black lines denote the initial configuration in each panel, red lines denote the deformed configuration. The table gives the conditions in the four panels, where a “+” force corresponds to stretching and a “-” force corresponds to compression. Growth is not incorporated here; size changes are due to rebalancing of the forces under the different conditions and result from elastic changes in the lengths of the edges.

**Figure 14 cancers-15-04840-f014:**
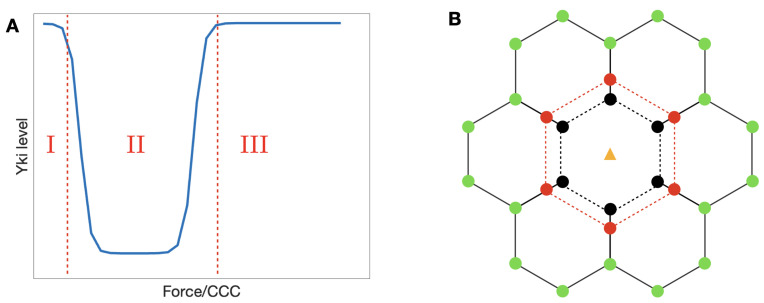
(**A**) A qualitative representation of the Yki level in response to tension exerted on α-catenin. (**B**) A simple framework of the growth model.

**Figure 15 cancers-15-04840-f015:**
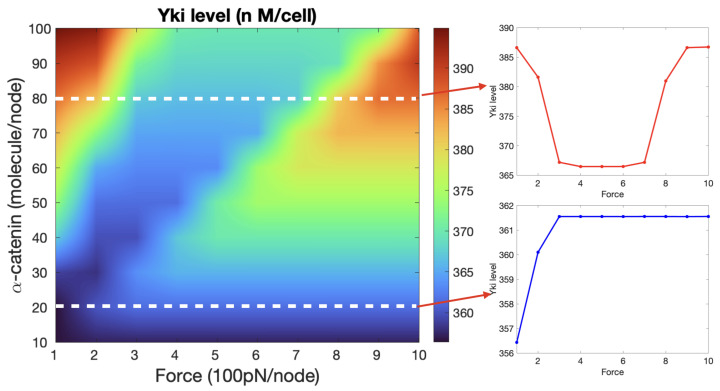
The heat map shows how the Yki level varies with different tension force applied on the CCC complex at each node and with the level of α-catenin at each node. The two plots on the right correspond to the two horizontal slices indicated by the white dashed lines shown in the heat map, where the level of α-catenin takes the value of 80 molecules/node (upper panel) and 20 molecules/node (lower panel).

**Figure 16 cancers-15-04840-f016:**
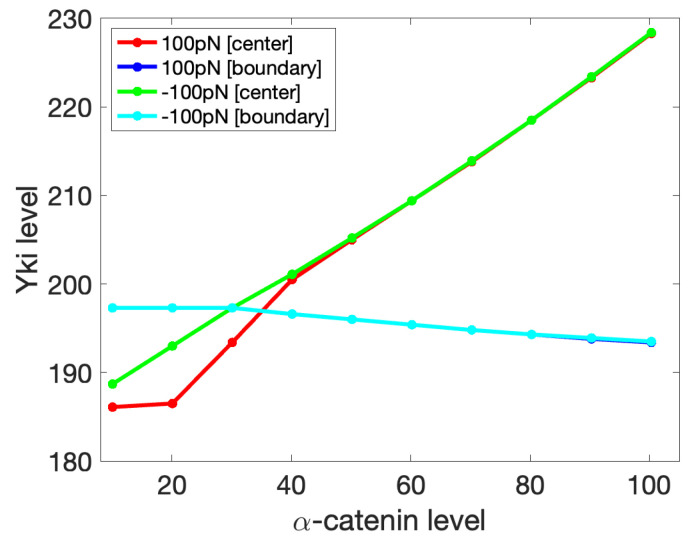
The Yki level (nM/cell) in the boundary cell ([boundary]) and center cell ([center]) as a response to a change in the α-catenin level (molecules/node) in the center cell, while the α-catenin level in the boundary cells is constant at 30 molecules/node. Shown are the results for the boundary forces of −100 pN (compression) and 100 pN (tension). The dark- and light-blue curves coincide.

**Figure 17 cancers-15-04840-f017:**
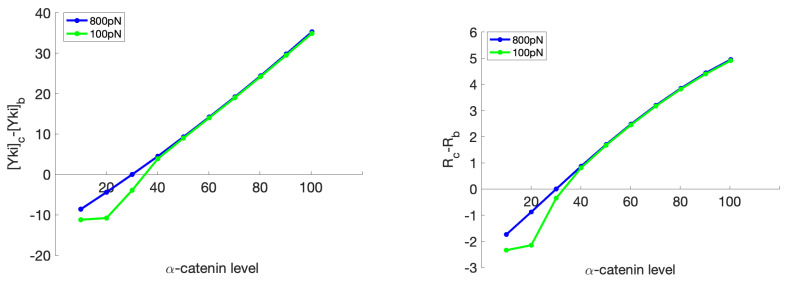
The difference in the Yki level (nM/cell) (left) and the ratio of change in the rest length (%) (right) between the boundary cells and the center cell as a response to a change in the α-catenin level (molecules/node) in the center cell, while the α-catenin level of the boundary cell is constant at 30 molecules/node. Here, the subscripts “c" and “b" indicate the center cell and a boundary cell, respectively, and R denotes the ratio of the final resting length to the initial resting length.

**Figure 18 cancers-15-04840-f018:**
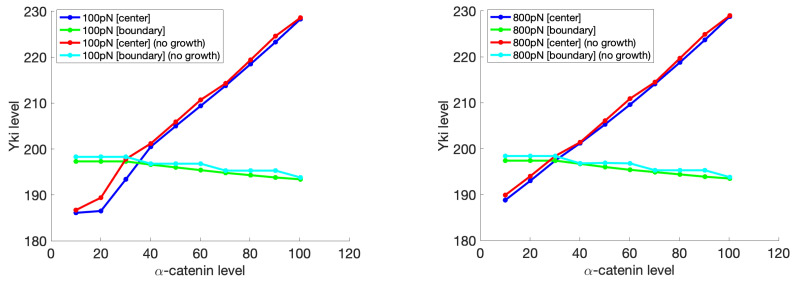
Comparison of the Yki levels as a function of the α-catenin levels with and without growth of the center cell under different environmental forces. Other settings are the same as those shown in Figure 17.

**Figure 19 cancers-15-04840-f019:**
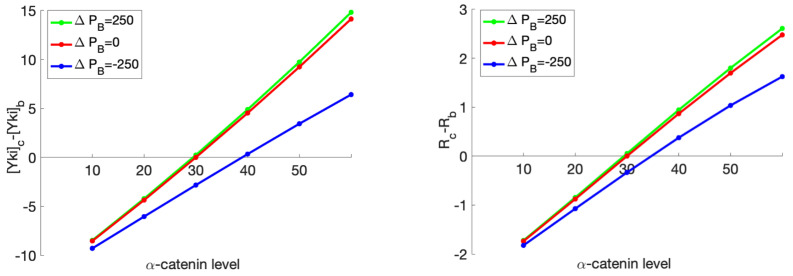
The difference in the Yki level (nM/cell) (left) and the ratio of change in the rest length (%) (right) between the boundary cells and the center cell as a response to a change in the production rate of Jub (nM/min) in the center cell, while the production rate of Jub for the boundary cell was set at a constant value of 250 nM/min.

## Data Availability

The data presented in this study are available on request from the corresponding author.

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
