# Peer review of "The Interaction of Mechanics and the Hippo Pathway in Drosophila melanogaster"

_cancers, 2023, doi:10.3390/cancers15194840_

Round 1

Reviewer 1 Report

The authors present a complex mathematical model that integrates the mechanical interactions between Drosophila cells with the intracellular actin network and the Hippo pathway to better understand cell growth in response to mechanical forces.

The article is well written, contains lots of biological explanations, the simulations are interesting.

The only downside is the fact that it is an extremely complex model, with lots and lots of parameters (some chosen randomly) and therefore this raises questions about the predictive values of the model and the numerical simulations. Hence, a sensitivity and uncertainty analysis would be useful. Also a discussion of the results in the light of uncertain parameter values

Author Response

We appreciate the reviewer's careful reading of the manuscript and his/her identification of the potential for a major weakness in large models of the type we formulated. Sensitivity analysis is needed for such models, and we previously did this for the model that the current paper is built on. There we identified 6-8 key parameters and we believe that a similar number will prevail here. We have added a long paragraph to the future directions portion of the Conclusion section in which we discuss this issue further and point out that a detailed sensitivity analysis is a project by itself, but carries with it the potential for simplifying the full model. 

Reviewer 2 Report

This is a very well-written paper focussing on an important problem in cell biology (with implications/applications to development, wound healing, and cancer among other topics). The introduction and background are clearly and concisely stated; the description of the general modelling approach is very well done (with details appropriately left to the Appendices); the model is parameterised appropriately; the model results/computational simulations are insightful and provide the basis for future work. I wish every paper I reviewed were as good as this one.

I have only one minor issue:

In Section 3.3, the authors describe the cell-cell network interactions via a Kelvin-Voigt model based on work in reference [98]. However, the work in [98] uses a linear viscoelastic model with power law creep compliance and notes (Supplementary Material) that neither a Maxwell nor a Kelvin 2-parameter model fitted their data well. Also, recently Villa et al. (2021) have shown that the constitutive equation can play an important role in pattern-formation in tissue-based systems. 

Villa, C., Chaplain, M.A.J., Gerisch, A., Lorenzi T. (2021) Mechanical models of pattern and form in biological tissues: The role of stress-strain constitutive equations. Bull. Math. Biol. 83, 80.

In a revised manuscript, perhaps the authors could provide some additional comments concerning the importance of constitutive equations.   

Minor typographical errors, lines 633 and 634/635:

[12] Shraiman, B.I. Mechanicall feedback... : Mechanicall --> Mechanical 

[13] Aegerter-Wilmsen, T ... <i>Drosophila</i>  :  remove the html brackets          

Author Response

We thank the reviewer for their careful reading of the manuscript and for pointing out what turned out to e an error in a citation. The reviewer noted that the author of the paper we cited had stated that the KV model was not appropriate for their system, but that citation was not the intended one! The citation we intended to use was to the paper by Iyer et al, now cited, in which they studied Drosophila wing disc tissue and stated explicitly that it adequately described their experimental results. We have corrected that error and included a reference to the paper by Villa, et al. referred to by the reviewer.